# *Bifidobacterium animalis* A12, a Probiotic Strain That Promotes Glucose and Lipid Metabolism, Improved the Texture and Aroma of the Fermented Sausage

**DOI:** 10.3390/foods12020336

**Published:** 2023-01-10

**Authors:** Yan Zhang, Yubing Hou, Shunliang Zhang, Nanqing Jing, Hongxing Zhang, Yuanhong Xie, Hui Liu, Jianguo Yan, Jianhua Ren, Junhua Jin

**Affiliations:** 1Beijing Laboratory of Food Quality and Safety, Beijing Key Laboratory of Detection and Control of Spoilage Organisms and Pesticide Residues in Agricultural Products, Food Science and Engineering College, Beijing University of Agriculture, Beijing 102206, China; 2China Meat Research Center, Beijing 100068, China; 3Key Ningxia Saishang Dairy Co., Ltd., Yinchuan 750299, China; 4Key College of Bioengineering, Beijing Polytechnic, Beijing 100176, China

**Keywords:** fermented sausage, *Bifidobacterium animalis* A12, hydrolysed amino acids, volatile flavour compounds

## Abstract

*Bifidobacterium animalis* A12 was used for the development of fermented sausage. The growth activity, tolerance, and enzyme activity of *B. animalis* A12 and its contribution to the texture and flavour of fermented sausages were evaluated. Additionally, the sensory texture, flavour components, and amino acid nutrients during the fermentation process were assessed. *B. animalis* had high tolerance to NaCl and nitrite, and *B. animalis* A12 had protease and lipase activities. The pH value of sausage fermented with *B. animalis* A12 was lower than that of sausage fermented without any fermentation strain. Hexanal, heptanal, decanal, cis-2-decanal, and 4-methoxy-benzaldehyde are the unique aldehydes flavour components of fermented sausages in the A12 group. The highest content of volatile flavour substances and amino acids, and the color and texture characteristics of fermented sausage in the experimental group at 18 h were better than those at other times. These results suggest that *B. animalis* A12 has the potential to be used as a starter culture for im-proving flavour and texture in fermented sausage.

## 1. Introduction

Fermented sausage is a meat product prepared by microbial fermentation and mixed with meat batter and ingredients. It has a long shelf life and a unique flavour. According to moisture content, fermented sausage is mainly divided into fermented dry sausage and fermented semi-dry sausage.

Traditional fermented sausage is mainly produced by natural fermentation, but the safety and small scale of the obtained products cannot be guaranteed. Therefore, to improve product quality and safety and further realise large-scale production, most researchers control the fermentation process by inoculating starters. Inoculation of appropriate probiotics in fermented sausages can prolong storage time and improve nutritional value and meat quality. The microorganisms in fermented sausage are mainly divided into lactic acid bacteria, including *Lactobacillus*, *Lactococcus*, *Pediococcus*, *Leuconostoc* [1], and *Weissella* [2]. The main role of the bacteria is to promote the rapid decline of pH, inhibit the adverse changes in raw meat caused by spoilage microorganisms or non-microbial reactions, stabilise the products, prolong the shelf life, and greatly improve the colour and taste of the products [3]. Other microorganisms in fermented sausage are *Staphylococci* and *Micrococcus*, most of which are *S. xylose* and *S. carnivores* [4]. By releasing bioactive substances such as lipase and protease, this kind of microorganism ultimately achieves the purpose of improving product colour and flavour. Yeast, which mainly exists on the surface of fermented sausage, can improve colour and produce a unique flavour [5], and can also consume oxygen in the environment, providing growth conditions for anaerobic or facultative anaerobic strains. Probiotics are defined as living microorganisms, mainly lactic acid bacteria (LAB) and *Bifidobacteria*, which can benefit host health when ingested properly. LAB, one of the most important varieties of fermented meat products, and one of the earliest and most widely used probiotics in food, have been widely used as starters to improve product quality and safety. Studies have demonstrated the feasibility of probiotics in fermented sausages. Gao et al. found that LAB C2 could effectively improve the sensory characteristics, colour, and flavour of fermented sausages [6]. Xi et al. [7] found that, compared with the control group, the nitrite content of sausages inoculated with *Lactobacillus plantarum* and *Lactobacillus ferment* decreased rapidly during maturation, which improved the safety of meat products. Cheng et al. inoculated *L. plantarum* and *Pediococcus pentosaceus* into sausage, and the results showed that the mixed starter had the potential to improve the quality of sausage [8]. Özer and Kılıç found that the addition of *L. plantarum* to fermented sausages could produce highly-conjugated linoleic acid [9]. 

Sidira et al. studied the effect of *Lactobacillus casei* addition on the formation of volatile compounds in fermented sausage production and found that esters and alcohols were high [10]. Hu et al. [11] detected 120 volatile compounds, including alcohols, acids, aldehydes, ketones, esters, and terpenes, in fermented sausages by Gas chromatography-mass spectrometry (GC-MS). Hu et al. [12] found that *P. pentosaceus*, *L. brevis*, *L. bentus,* and *L. fermentans* isolated from harbin air-dried sausages were precursors for the decomposition of sarcoplasmic proteins to form flavour compounds. Although a large number of probiotics have been reported for fermented sausage, the application of *Bifidobacterium animalis* alone has not been reported. Therefore, the present study expands on current knowledge and highlights new techniques for meat processing.

*B. animalis* A12 can inhibit the production of glucose and hinder the transport of glucose, thereby reducing blood glucose levels and exerting a potential hypoglycaemic effect [13]. In addition, *B. animalis* A12 can effectively inhibit obesity induced by a high-fat diet [14]. Therefore, it is of great significance to apply *B. animalis* A12 to fermented sausages as a nutritional functional food.

In this study, the tolerance evaluation and enzyme production capacity of the experimental strains were first carried out, and then applied to fermented sausage to evaluate the sensory, textural, and hydrolytic amino acids and volatile flavour components of fermented sausage. This provides a basis for the development of functional foods using probiotics.

## 2. Materials and Methods

### 2.1. Strains and Culture

*P. pentosaceus* S2, *Staphylococcus vitulinus* T1, and *B. animalis* A12 (CGMCC No.17308) were inoculated into MRS and nutrient liquid medium for amplification and culture to the third generation.

### 2.2. Growth and Acid Production Capacity of the Strain

The activated strains S2, T1, and A12 were inoculated in sterilised 200 mL MRS and nutrient liquid medium at 1% proportion, respectively, and cultured on a shaking table at 37 °C and 200 rpm. Using sterile medium as the control, the samples were collected at 0, 2, 4, 6, 8, 10, 12, 14, 16, 18, 20, 22, and 24 h of culture, and the OD600 value and pH value of the fermentation broth were determined.

### 2.3. Salt Tolerance and Nitrite Tolerance of Strains

Different concentrations of sodium chloride (0, 2.5, 5, 7.5, and 10%) and nitrite (0, 50, 100, 150, and 200 mg/L) were added to the liquid MRS and nutrient medium, respectively, for sterilisation. The activated strains S2, T1, and A12 were inoculated in MRS and nutrient liquid medium containing different concentrations of sodium chloride and sodium nitrite and cultured in a constant temperature incubator at 37 °C for 24 h. The OD600 value was determined.

### 2.4. Protease Production Capacity of Strains

The activated strains S2, T1, and A12 were inoculated into the PCA solid medium supplemented with 2% casein at 200 μL according to the Oxford cup method. Sterile water was used as a control, and it was placed in 37 °C constant temperature incubator for 5 days. Plastic droppers were used to put 3–5 drops of 10% trichloroacetic acid in the culture medium around the colony drops and the size of the transparent circle was measured.

### 2.5. The Lipase-Producing Ability of the Strains

The activated strains S2, T1, and A12 were inoculated on PCA solid medium supplemented with 12% lard oil and 1 mL 1.6% bromocresol purple solution according to the Oxford cup method. A transparent circle was observed when sterile water was added as a control and the cells were cultured at 37 °C for 5 days.

### 2.6. Fermented Sausage Technology

The experimental raw materials (pork meat (85%), back fat (15%)) and excipients (the proportion of experimental excipients to the following experimental raw materials—carrageenan (1%), soy protein isolate (2%), iso-Vc-Na (0.05%), sodium tripolyphosphate (0.3%), sodium nitrite (0.003%), salt (1.8%), sugar (0.2%), glucose (1%), ice water (30%), monascus red (0.1%).)—were mixed in proportion, the pre-treated bacterial suspension was inoculated into the A12 and experimental group, and the mixture was evenly mixed, after enema, fermentation, fermented sausage at different times was taken for index detection. The entire experiment was repeated thrice.

CK group: no bacterial inoculation and natural fermentation; A12 group: adding *B. animalis* A12 freeze-dried powder (9 × 10^11^ CFU/kg), fermentation temperature (40 ℃), humidity (95%), time (18 h); experimental group: *P. pentosaceus* S2, *Staphylococcus vitulinus* T1, and *B. animalis* A12 (1:3:1) were inoculated. The inoculation amount was 4 × 107 cfu/g, Sampling at (0 h: T0; 6 h: T6; 12 h: T12; 18 h: T18; 24 h: T24).

### 2.7. Sensory Score

Ten students who had studied sensory evaluation were selected as judges to score the appearance (X1), organisation (X2) and flavour (X3) of the fermented sausage (Table 1). Before sensory evaluation, the judges washed their hands and maintained oral hygiene. Each sample was evaluated once and each score was denoted as Xi (i = 1, 2, 3). The total score was X = 0.2 × 1 + 0.3 × 2 + 0.5 × 3 and the final score was the average of the sum of each X.

### 2.8. Determination of pH, Colour, and Texture of Fermented Sausage

According to GB 5009.237-2016, a calibrated pH meter (SARTORIUS, PB-10, Goettingen, Germany) was inserted into the sausage to determine the pH value.

The colour was measured with a colorimeter (KCM-700d, ONICA MINOLTA, Tokyo, Japan) using an 8 mm port size, illuminant D65, and 10° standard observer. The *L** (lightness), *a** (red) and *b** (yellow) value of fermented sausage were measured and recorded after whiteboard correction. There were 3 parallel samples for each sample, 5 points for each parallel sample, a total of 15 measured values, and the average of 15 measured values was used as the analytical standard.

The texture was measured with a texture analyser (Stable Micro Systems, TA-XT Plus, Surrey, UK), which was repeated three times and the average value was taken. Using the texture profile analysis (TPA) method, the P50 cylindrical probe was used to cut the fermented sausage into samples with a diameter of 12.7 mm and a length of 20.0 mm. The compression rate was 2.0 mm/s before the test, and 1.0 mm/s during the test. After testing, the compression rate was 1.0 mm/s, the compression ratio was 75%, and the pause time between the two compressions was 5.2 s, and the following textural parameters were calculated: hardness (N), chewiness (N* mm), and elasticity (mm).

### 2.9. Determination of Hydrolysed Amino Acids

An amount of 10.0 g of fermented sausage was weighed and dissolved in 10 mL of 6 mol/L HCl. Phenol was added and frozen and vacuum filled with nitrogen seal. It was cooled by hydrolysis. The hydrolysate was filtered through medium-speed qualitative filter paper and the filtrate was dried. Finally, pH 2.2 sodium citrate buffer was added to fully dissolve the samples, and after mixing, the samples were filtered through a 0.22 μm filter membrane. Finally, the amino acid analyzer was used for determination and analysis. Instrument conditions: analytical column: 4.6 mm × 60 mm; resin: 2622#; column temperature: 57 °C; reaction column temperature: 135 °C; buffer: citric acid, sodium citrate buffer; chromogenic solution: ninhydrin solution.

### 2.10. Determination of Volatile Flavour Components in Fermented Sausage

The volatile flavour components of fermented sausage were determined by GC-MS. Accurately weighed 10.0 g cut sausage samples were added to the sample bottle, 1 μL of 0.816 μg/μL 2-methyl-3-heptanone was added as the internal standard, and the bottle was closed. After the sample bottle was preheated in a 55 °C water bath for 10 min, an SPME needle was inserted into the sample bottle, and the fibre head was placed on top of the sample bottle. After the volatile flavour substances in the sample bottle were adsorbed for 40 min, they were removed and inserted into the GC inlet for desorption for 10 min. GC conditions: TG-Wax MS polar column (30 m × 0.25 mm, 0.25 μm); the carrier gas was high purity helium (purity > 99.99%); flow rate: 1.5 mL/min; use no shunt mode, keep 2 min. The temperature program was as follows: inlet temperature 250 °C; column temperature 40 °C; 3 min, 5 °C/min rate to 200 °C; maintained for 1 min; and finally, 8 °C/min rate to 230 °C, and maintained for 10 min. The MS conditions were as follows: transmission line temperature, 230 °C; electron energy, 70 eV; electron ion source temperature, 280 °C; mass scanning range 40–600 u; full scan mode.

### 2.11. Statistical Analyses

Experimental data were expressed as mean ± standard deviation (SD). All experiments were repeated three times. SPSS 22.0 software (IBM, Chicago, USA) was used to analyze the data by independent sample T test to determine significant differences. Statistical significance was set at *p* < 0.05. GraphPad (GraphPad Software, Santiago, USA) was used to analyze the GC-MS data and construct a heatmap.

## 3. Results

### 3.1. Study on Growth and Tolerance of Strains

The growth curve is an intuitive reflection of the growth vigour of the strain. As shown in Figure 1A, the OD600 values of strains S2, T1, and A12 changed slightly at 0–2 h during the incubation period. After 2 h, the three strains entered an exponential growth period, and the OD600 values increased rapidly. Strain T1 entered a stable period at 6 h, and strains S2 and A12 entered a stable period at 8 h. At the stable stage, all three strains showed good growth vigour.

The pH value is an important index of raw material carbohydrate decomposition, which can inhibit the growth of pathogenic microorganisms and improve the stability of the product. As shown in Figure 1B, the pH values of the three strains generally showed a downward trend and tended to stabilise at 12 h. At 0–12 h, the pH values of strains S2 and A12 decreased from 6.76 and 6.62 to 4.78 and 4.49, respectively. During the entire culture process, the pH value of strain A12 decreased significantly, and was always lower than that of T1 and S2, indicating that strain A12 had a stronger acid production ability. The acidity of fermented meat products should be maintained at the isoelectric point of the muscle protein (pH 5.2) to effectively inhibit the growth of spoilage microorganisms. Therefore, the pH value in fermented meat should be maintained at 5.2 to ensure that the comprehensive quality of sausage is the best.

As shown in Figure 1C, the OD600 of the three strains decreased with increasing NaCl concentrations. When the NaCl concentration was 0–2.5%, all three strains had high OD600, the NaCl concentration was 5%, and the OD600 of S2 and T1 significantly decreased (*p* < 0.05). S2 maintained a high OD600 at a NaCl concentration of 7.5–10%, indicating that the strain had high tolerance to NaCl. It has been reported in the literature that 2.0% NaCl is the ideal addition amount of fermented sausage [15]. *B. animalis* A12 has good growth ability within the concentration range of the commercial starter strains S2 and T1, indicating that *B. animalis* A12 is suitable for the fermentation of meat products.

Nitrite is a commonly used food additive in meat products, which has colour development and antibacterial effects and helps to improve meat quality [16]. According to the Chinese national standard GB5009.33-2016 regulation, the addition of sodium nitrite to fermented meat products is less than 150 mg/kg. In this study, according to actual needs, the addition amount was 30 mg/kg, as shown in Figure 1D. At this time, the nitrite tolerance of *B. animalis* A12 was similar to that of commercial starter culture, and all three strains had high OD600 values, indicating that *Bifidobacterium animalis* A12 could adapt to the nitrite content in the experiment and maintain good growth ability, which could be used as meat product starter culture.

### 3.2. Enzyme Production Capacity of Strains

The lipase and protease production results for the three strains are shown in Table 2. Strains S2 and A12 exhibited good lipase production capacity. Strain S2 had a stronger lipase production capacity than strain A12, whereas strain T1 did not produce lipase. All three strains had protease production capacities, and strain T1 had the highest protease production capacity.

### 3.3. Effect of B. animalis A12 on Sensory, Color and Texture of Fermented Sausage

As shown in Table 3, the sensory scores of the CK and A12 groups were significantly different (7.12 vs. 7.67, *p* < 0.05). The sensory quality of sausages fermented by A12 was significantly improved, with a unique flavour. This indicated that *B. animalis* A12 played an important role in the sensory quality of fermented sausages. After 18 h of fermentation, the pH of A12 group decreased significantly (5.048 vs. 5.927, *p* < 0.05). Previous research results showed that after inoculation with *L. plantarum* and simulated S. aureus, pH decreased, which was similar to our results [17]. Therefore, the addition of A12 effectively reduced the pH of the fermented sausages. Low pH can inhibit the growth and reproduction of spoilage microorganisms and cause protein denaturation in meat, thereby improving the quality and safety of sausage [18]. At the same time, the *L** value (brightness, 68.41 vs. 63.03, *p* < 0.05) and a* value (red, 8.12 vs. 6.69, *p* < 0.05) of the A12 group were significantly higher than those of the CK group, which may be because the metabolites of A12 have reducibility and improve the colour of the sausage. Eduardo et al. [19] confirmed that the colour of sausage with probiotics was redder than that without probiotics and had better sensory quality. *B. animalis* A12 could effectively improve the colour of sausage from the perspective of comprehensive quality, which might be due to the low pH value of fermented sausage by *B. animalis* A12, which promotes the decomposition of nitrite to NO and then generates nitroso myoglobin, thereby improving the colour of the product [20,21]. The hardness (1292.421 vs. 2202.62 N, *p* < 0.05) and chewiness (743.23 vs. 1216.096 N*mm, *p* < 0.05) of the A12 fermented sausage were significantly lower than those of the CK group (*p* < 0.05), which was consistent with the experimental results of Kanno et al. [22]. This may be due to the low pH, which reduces the water-holding capacity of proteins in sausages. The elasticity of A12 fermented sausage was significantly higher than that of the CK group (0.794 vs. 0.725, *p* < 0.05). Essid and Hassouna found that inoculation with *Staphylococcus xylosus* and *L. plantarum* could increase the elasticity of fermented sausages [23], which is consistent with our results. The increase in elasticity during sausage maturation is mainly due to dehydration. In summary, *B. animalis* A12 can effectively improve the texture of fermented sausage, indicating that it can be used as a sausage starter.

### 3.4. Effect of B. animalis A12 on Volatile Flavour Compounds of Fermented Sausage

Volatile flavour compounds in the two fermented sausages are shown in Figure 2 and Appendix A. Detected in two groups of fermented sausages mainly include aldehydes, alcohols, ketones, acids, esters and others, a total of 52 kinds. There were 16 types of volatile flavour compounds in the CK group and 46 in the A12 group. Those in A12 group were significantly higher than those in the CK group. The aldehydes, alcohols, ketones, acids, and esters in the A12 group contained 15, 7, 4, 4, and 7, respectively.

The proportion of aldehydes was the highest in these two groups. The content of aldehydes in the A12 group accounted for approximately 62% of the total volatile components, and its significantly higher than that in the CK group (*p* < 0.05). This indicated that the main flavour components in the sausage were aldehydes, followed by acids and alcohols. Among them, nonanal is the most abundant and important volatile flavour component in A12 fermented sausage, which is described as a lemon and plastic flavour [24]. At the same time, hexanal, heptanal, decanal, cis-2-decanal, and 4-methoxy-benzaldehyde were detected in the A12 group, which were not found in CK group. Heptanal and decanal have fruit, cucumber, and liquorice flavour [25,26], respectively, indicating that *B. animalis* A12 can significantly improve the flavour of fermented sausage.

Alcohols have a high detection threshold and have less of an effect on meat aroma than aldehydes [27], but it also plays a key role in the flavour of fermented sausage [28]. Among them, 1-octene-3-ol, a degradation product of linoleic acid hydroperoxide [29], has a mushroom flavour [30] and has been reported to enhance the flavour of meat. Therefore, it is a typical meat flavour compound. At the same time, it is also a unique alcohol substance that was found in A12 group; therefore, it plays an important role in the flavour of fermented sausage. In the CK group, only 1-heptatriol was detected.

Ketones contribute to the production of milk flavours. 2-nonanone, acetoin, and hydroxyacetone are unique volatile flavour components in the A12 group. It has been reported that acetoin, which has a butter and cheese flavour, significantly contributes to the flavour of sausage [31].

Acids are mainly produced by the fermentation of carbohydrates by LAB and degradation of branched-chain amino acids. Acetic acid is a unique flavour substance of the fermented sausage in the A12 group.

It has been reported that esters are produced by the esterification of alcohols and acids [32,33], and most of them have an aromatic flavour. The esters formed by short-chain acids are mostly fruity, and those formed by long-chain acids are mostly light oil flavours. Therefore, esters play a role in flavour of fermented sausages. The above results showed that *B. animalis* A12 plays an important role in the flavour formation of fermented sausage.

### 3.5. Effect of B. animalis A12 Mixed Starter on pH, Colour, and Texture of Fermented Sausage

As shown in Table 4, the pH during the maturation process of fermented sausage generally showed a downward trend, indicating that the added starter strain had a certain ability to produce acid. During 0–12 h of initial fermentation, the pH of fermented sausage showed a significant downward trend, indicating that the activity of the starter was high, which could rapidly reduce the pH value of meat products and help produce a good flavour. Colour is an important factor that reflects the quality of fermented sausage. The *L*, a**, and *b** values of each time period showed no significant changes during the fermentation process, indicating that the fermentation strain had no obvious effect on the colour of sausage and could effectively maintain the original colour of the meat. The value of hardness, elasticity, and chewiness of fermented sausage increased first and then decreased, and were significantly higher at 18 h than those at the other time points.

### 3.6. Effect of B. animalis Mixed Starter on Amino Acids in Fermented Sausage

The contents of hydrolysed amino acids in the sausages at different fermentation times are shown in Figure 3 and Appendix A. The total amino acid content in sausages at 0 h was 782.43 mg/kg dry matter. With the extension of the fermentation time, the amount of various amino acids first increased and then decreased. This indicated that during the fermentation and maturation of sausage, macromolecular proteins and amino acids released by peptide hydrolysis further reacted to produce other small-molecule substances. These amino acids may be produced by protein hydrolysis and are dominant in the late ripening process, which contributes to the formation of the probiotic fermented sausage flavour. At 18 h, the total amount of amino acids was up to 1015.63 mg/kg dry matter. At this time, the total amount of essential amino acids and non-essential amino acids in sausage was greater than that at other fermentation times, mainly because of the effect of microorganisms on protein. Under the action of peptidase, the protein decomposes to produce polypeptides and various amino acids. The total amino acid content at 18 h was significantly higher than that at 0 h (*p* < 0.05). The results showed that the inoculation starter had good aminopeptidase activity and could significantly improve the amino acid content in fermented sausages. The total amount of essential amino acids was 469.2 mg/kg, accounting for 46.2% of the total amino acid content, while the total amount of non-essential amino acids was 546.44 mg/kg, accounting for 53.8% of the total amino acid content. Asp and Glu are considered delicious amino acids that contribute to the umami taste of food [34]. At 18 h, it accounted for 15.9% of the total amino acid content. After 18 h, the total amount of hydrolysed amino acids decreased. Therefore, the above results suggest that *B. animalis* A12 plays a positive role in the amino acid content of fermented sausages, which is most obvious at 18 h.

### 3.7. Effect of B. animalis Mixed Starter on Volatile Flavour Compounds of Fermented Sausage

Figure 4 and Appendix A showed that 13, 42, 53, 53, and 52 volatile flavour compounds, including 10 common components, were detected in T0, T6, T12, T18, and T24 fermented sausage samples. The total content of volatile flavour compounds was 179.67, 233.43, 512.45, 627.47, and 443.02 μg/kg, respectively. The content of volatile flavour compounds in fermented sausages first increased and then decreased. The content of volatile flavour compounds in T18 was significantly higher than that in the other time (*p* < 0.05), indicating that the flavour compounds of sausage were directly related to fermentation time. Among the five time points of fermented sausage, aldehydes were the most abundant, with 4, 16, 20, 20, and 20 species detected, with total contents of 95.23, 160.89, 340.30, 416.56, and 284.76 μg/kg, respectively. The proportions were 53.00, 68.92, 66.41, 66.39, and 64.28%, respectively, which indicated that the flavour substances in sausage produced more flavour substances in the fermentation process, indicating that the processing technology of sausage was helpful to promote the formation of flavour substances in sausage. Among them, the contents of n-octanal, nonanal, hexanal, and heptanal, which are the main components of sausage flavour, increase with the extension of fermentation time and reach the highest value at 12–18 h. Aldehydes are generally produced by degradation or oxidation of fatty acids. Owing to their low perception threshold, they become the key substances that constitute the characteristic flavour of meat [35].

Alcohols have less significant effects on meat aroma than aldehydes, but they also play a key role in the overall flavour [36]. Alcohols were not detected in T0, which was due to the low alcohol content of the sausage itself. At the same time, with the extension of fermentation time, the alcohol content in the sausage first increased and then decreased. Alcohols were detected in the T6, T12, T18, and T24 (6, 8, 8, and 8, respectively), and the total contents were 9.89, 21.79, 35.35, and 24.87 μg/kg, accounting for 4.23, 4.25, 5.63, and 5.61%, respectively. 

The total ketone content in sausage first increased and then decreased with time. Ketone compounds are related to citric acid and lactose metabolism and can also be produced through amino acid decomposition [37]. Methylheptenone was detected in the T6, T12, T18, and T24, which can add lemongrass and mint aromas to sausage [38].

The sausage samples also contained small amounts of acids, esters, alkanes, and other compounds. 2-pentylfuran is a heterocyclic compound with the greatest impact on flavour and is mainly derived from fat oxidation.

## 4. Discussion

Current research has shown that the application of probiotics in sausage fermentation is beneficial for improving the quality and nutritional value [39,40]. Therefore, this study aimed to evaluate the effects of *B. animalis* A12 on pH, colour, texture, hydrolysis of amino acids, and volatile flavour components of fermented sausages.

During the fermentation process, some basic indicators change, which determines the quality of the final product. These changes are mainly caused by fermentation acidification, pH reduction, nitrate reduction to nitrite and nitroso myoglobin formation, proteolysis, lipolysis, and oxidation [41,42]. LAB are a dominant flora in the production of fermented sausage, and play a very important role in reducing the pH value of sausage, which is an important indicator of raw material carbohydrate decomposition. In a low-pH environment, the growth of some adverse bacteria and pathogenic microorganisms can be inhibited, which is helpful for improving the stability of the products. Our results showed that the pH value of fermented sausage in the A12 group was lower than that in the CK group in 18 h, which may be due to the lactic acid produced by *Bifidobacterium*, similar to the results reported in the literature [43,44,45,46,47,48,49,50,51,52,53,54,55]. The NaCl concentration in fermented sausage is generally higher; therefore, it is necessary to choose a salt-tolerant starter.

To grow the strain during fermentation and maturation, it is important to have tolerance to acids, salts, and nitrite [46]. The production of organic acids is an important factor for ensuring the shelf life and safety of fermented sausages. The inhibition of pathogenic microorganisms depends on a decrease in pH [47]. The addition of salt to fermented sausages usually increases the taste of the product and prolongs its shelf life [48], while nitrite affects colour development and antibacterial properties [49]. Strains producing lipase and protease play a crucial role in the flavour of fermented sausage [50,51]. Evaluation of the tolerance and enzyme production ability of the three probiotics showed that the three strains have a strong ability to produce acid, protease, and lipase, and a resistance to salt and nitrite, to meet the needs of fermented meat-processing technology. The results indicated that three strains can be used as meat starters.

During the processing and storage of fermented sausages, proteins are gradually degraded into peptides, small peptides, and amino acids. Amino acids are the final product of protein degradation, and their content and composition ratio play an important role in improving the taste of fermented sausages. Gly, Ala, Ser, and Pro are known to be sweet, while Asp and Glu are umami. Studies have shown that with the increase in microbial enzyme activity in the late fermentation of fermented sausage, the content of amino acids in the mature process increased, resulting in the significant release of amino acids [52,53], which is similar to our results. At T18, the essential amino acid content was significantly higher than that at T0 and T6, these results indicate that *B. animalis* A12 can promote amino acid production and accelerate protein breakdown in fermented sausages.

The volatile flavour compounds of the fermented sausages were analysed using GC-MS. Some studies have shown that probiotics have a positive effect on the formation of volatile flavours in sausage [54,55]. The main volatile compounds in fermented sausages are aldehydes, alcohols, ketones, acids, and esters. Aldehydes are primarily derived from lipid oxidation and amino acid decomposition. Because of their low thresholds, aldehydes have an important impact on the flavour of fermented sausages [56], therefore, they become the key substances that constitute the characteristic flavour of meat [57]. Hexanal, heptanal, decanal, cis-2-decanal, and 4-methoxy-benzaldehyde are the unique aldehydes flavour components of fermented sausages in the A12 group, which was rarely reported in previous similar studies. Nonanal is the main source of sweet orange aromas in the food industry. Alcohols have a less obvious effects on meat aroma than aldehydes, but they also play a key role in the overall flavour [58]. During sausage fermentation, alcohols are produced through various metabolic carbohydrate pathways, such as lactic acid metabolism and transformation, with the participation of probiotics [29,59]. Alcohol content indicates oxidative degradation of intramuscular fat and unsaturated fatty acids. Alcohols were not detected in the T0, which may be due to the lower alcohols in the sausage itself and the volatilisation loss of alcohols during cooking. Alcohols in sausages mainly include n-octanol, 1-octen-3-ol, and ethanol. 1-octene-3-ol, has a mushroom flavour and can enhance the flavour of meat. It is a typical meat flavour compound [60] and is the main contributor to the sausage flavour. Ethanol is the main compound produced by LAB metabolites [61]. Ketone compounds are produced by the thermal oxidation of unsaturated fatty acids and amino acids in pork [62]. At the same time, it has also been reported that the acetic acid of probiotic sausage increased significantly after fermentation [63]. This result is similar to our experimental results. This is because the presence of probiotics increases the acetic acid content caused by the fermentation process of carbohydrates in meat products, resulting in a decrease in pH. A decrease in pH can inhibit the growth and proliferation of adverse microorganisms, thereby extending the shelf life. There were also significant differences in ester content during fermentation. Esters are aromatic compounds that can produce fruit flavour [64], and they are also considered to have significant effects on the volatile flavour of fermented sausage. The sausage samples also contained small amounts of olefins, alkanes, and other substances. 2-pentylfuran is a heterocyclic compound with the greatest impact on flavour and is mainly derived from fat oxidation. Its green bean and butter flavours are considered important flavour substances in meat products [65].

Previous studies have reported that *B. animalis* cannot be used alone in fermented sausage, and that it must be combined with other probiotics [66]. In this study, when *B. animalis* A12 was used alone, the sensory quality of fermented sausage was improved. In addition, the hydrolysed amino acids and volatile flavour components were enhanced, indicating that there may be differences between strains. Therefore, it is necessary to strengthen this discussion further in the future.

## 5. Conclusions

Our findings showed that the application of *B. animalis* A12 in fermented sausage had a positive effect. First, the tolerance evaluation of *B. animalis* A12 showed that it exhibited acid resistance, salt resistance, nitrite resistance, and enzyme production. At the same time, the sensory score, colour, and texture were improved when *B. animalis* A12 was applied to fermented sausage, and the content of hydrolysed amino acids and volatile flavour components was highest in sausage fermented for 18 h. Hexanal, heptanal, decanal, cis-2-decanal, and 4-methoxy-benzaldehyde are the unique aldehydes flavour components of fermented sausages in the A12 group. Overall, the addition of *B. animalis* A12 improved the quality of fermented sausages; it can be used as a potential probiotic auxiliary culture for the development of functional fermented sausage.

## Figures and Tables

**Figure 1 foods-12-00336-f001:**
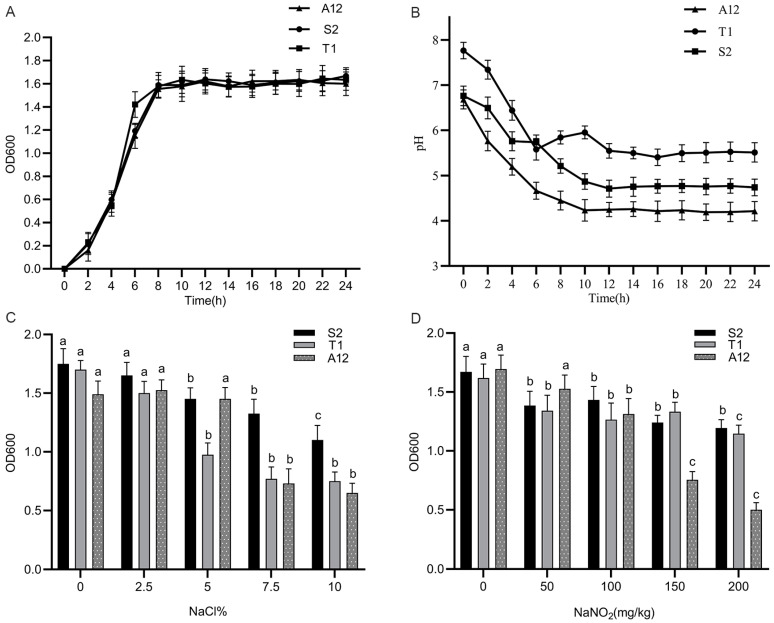
Evaluation of the growth and tolerance of the experimental strains. (**A**): growth curve; (**B**): pH; (**C**): salt tolerance; (**D**): nitrite resistance. a,b,c: different letters indicate significant differences in different salt and nitrite concentration environments in the same strain (*p* < 0.05).

**Figure 2 foods-12-00336-f002:**
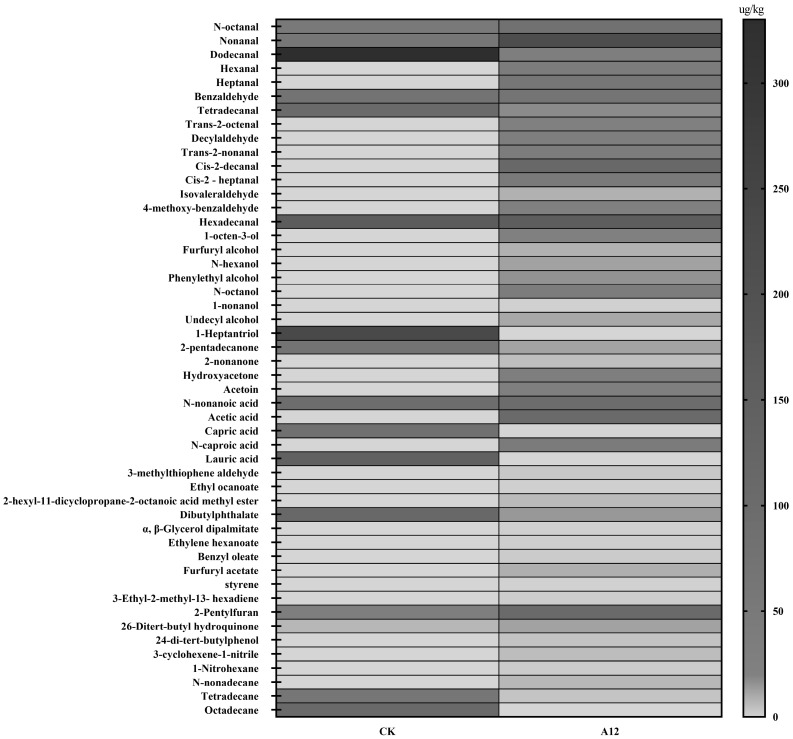
Effect of *Bifidobacterium animalis* on volatile flavour components of fermented sausage.

**Figure 3 foods-12-00336-f003:**
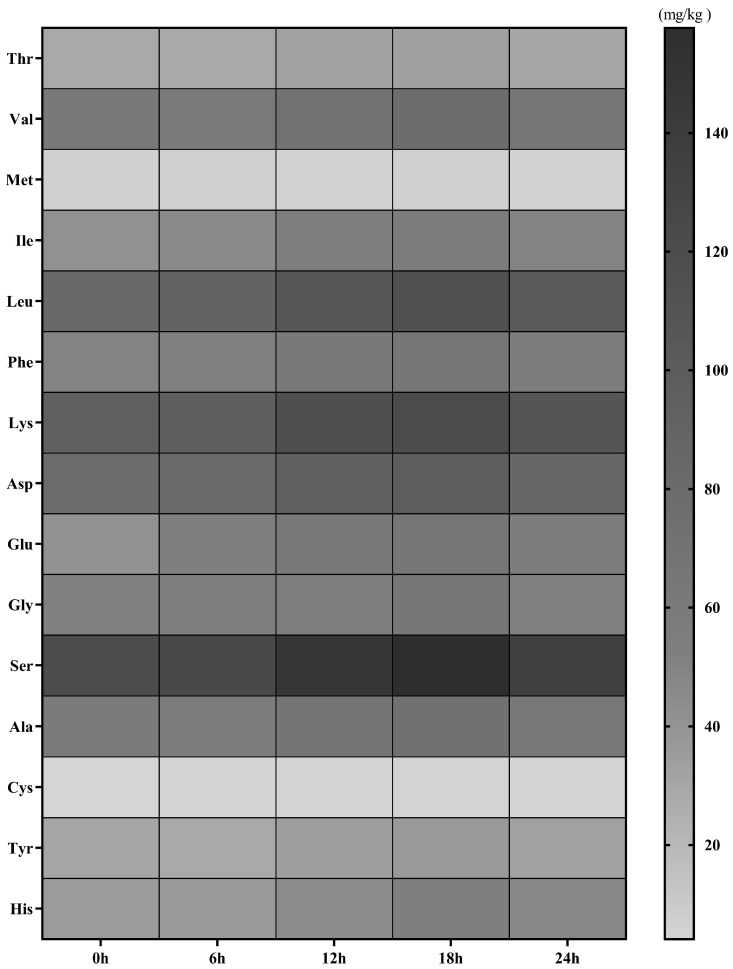
Heatmap of amino acid content in fermented sausage at different times.

**Figure 4 foods-12-00336-f004:**
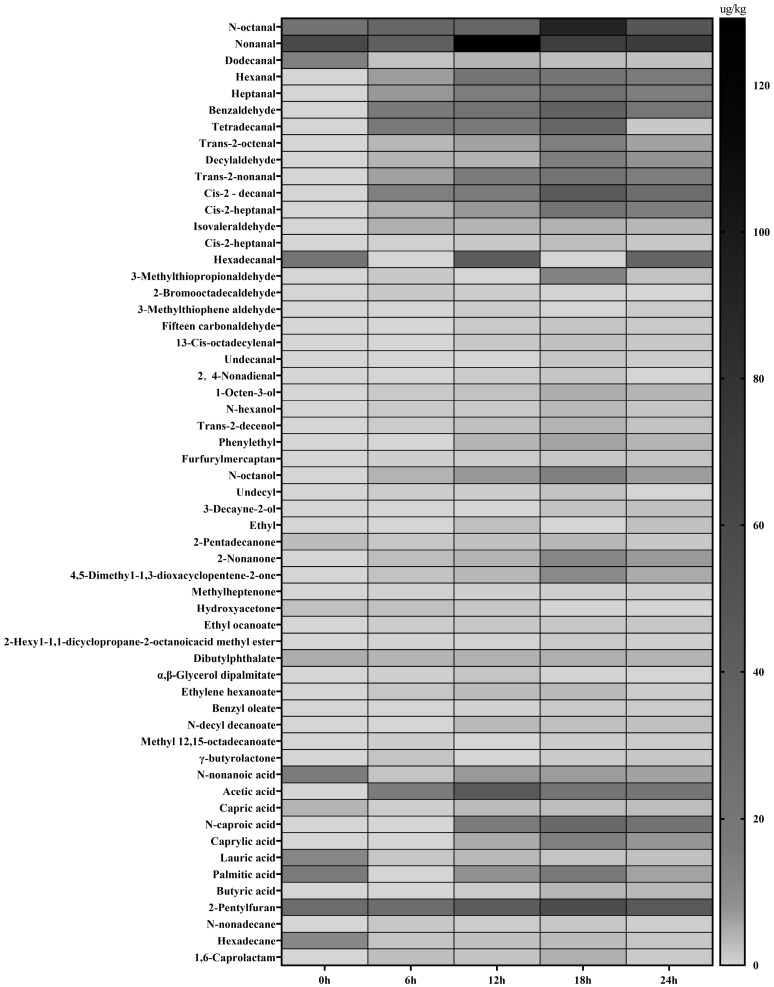
Heatmap of volatile flavour components in fermented sausage at different times.

**Table 1 foods-12-00336-t001:** Sensory evaluation of fermented sausage.

Indicator	Scores			
	9–10	6–8	3–5	0–2
**Appearance (0.2)**	Full bowel, close to the bowel	Fuller, better fit	Loose intestines, large gaps	Intestinal body loose, intestinal body and intestinal clothing separated
**Organisation (0.3)**	Good elasticity, chewability, neat section	Good elasticity, chewability, cracks	Elastic, poor chewing, cracked	Poor elasticity, poor chewiness, irregular section and cracks
**Flavour (0.5)**	Fermentative fragrance, tasteless, delicious	Fermentation fragrance is not obvious, no smell, taste better	Fermentation flavor is not obvious, smelly, taste general	No fermented fragrance, smelly, bad taste

**Table 2 foods-12-00336-t002:** Results of lipase and protease producing ability of strains.

Strain	S2	T1	A12
Fat PCA medium	+++	-	+
Casein PCA medium	++	+++	+

- incompetence; + weak ability (d < 0.5 cm); ++ medium capacity (1 cm < d < 1.5 cm); +++ strong ability (d > 1.5 cm).

**Table 3 foods-12-00336-t003:** Sensory, colour, and texture traits of fermented sausage in different treatment groups.

Item	Treatment	
	CK	A 12
Sensory score	7.12 ± 0.27 a	7.67 ± 0.13 b
pH (0 h) pH (18 h)	6.304 ± 0.192 a 5.927 ± 0.219 a	5.991 ± 0.219 a 5.048 ± 0.259 b
Color traits (18 h)		
*L**	63.03 ± 1.35 a	68.41 ± 1.12 b
*a**	8.12 ± 0.26 a	9.69 ± 0.34 b
*b**	14.97 ± 0.66 a	15.16 ± 0.54 a
Texture traits (18 h)		
Hardness (N)	2206.62 ± 96.37 a	1292.421 ± 72.66 b
Chewiness (N* mm)	1216.096 ± 55.41 a	743.23 ± 62.36 b
Elasticity (mm)	0.725 ± 0.03 a	0.794 ± 0.01 b

a,b: different letters indicate significant differences in different group. (*p* < 0.05).

**Table 4 foods-12-00336-t004:** pH, colour, and texture of fermented sausage at different time points.

Item	Time				
	0 h	6 h	12 h	18 h	24 h
pH	6.23 ± 0.094 a	5.78 ± 0.075 a	5.24 ± 0.075 a	5.06 ± 0.074 a	4.87 ± 0.077 a
Color traits					
*L**	68.65 ± 1.16 a	67.80 ± 1.34 a	68.44 ± 1.23 a	68.40 ± 1.12 a	67.94 ± 1.01 a
*a**	9.56 ± 0.31 a	10.61 ± 0.72 a	10.47 ± 0.52 a	9.94 ± 0.44 a	9.84 ± 0.38 a
*b**	13.64 ± 0.67 a	14.72 ± 0.32 a	14.06 ± 0.57 a	13.26 ± 0.66 a	13.41 ± 0.73 a
Texture					
Hardness (N)	2650.52 ± 420.61 b	2747.36 ± 361.47 b	2807.58 ± 267.81 b	3323.76 ± 314.27 a	2817.96 ± 234.17 b
Chewiness (N* mm)	0.73 ± 0.01 b	0.75 ± 0.03 b	0.81 ± 0.02 b	0.85 ± 0.02 a	0.73 ± 0.02 b
Elasticity (mm)	464.32 ± 20.53 d	667.15 ± 23.66 c	838.61 ± 33.26 b	964.32 ± 47.12 a	637.12 ± 51.63 c

a,b,c,d means in the same row with different letters are significantly different (*p* < 0.05).

## Data Availability

The data are available from the corresponding author.

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
