# Peer review of "Bifidobacterium animalis A12, a Probiotic Strain That Promotes Glucose and Lipid Metabolism, Improved the Texture and Aroma of the Fermented Sausage"

_foods, 2023, doi:10.3390/foods12020336_

Round 1
Reviewer 1 Report
Manuscript ID: Foods-2117587
In the article entitled: “Bifidobacterium animalis A12, a probiotic strain that promotes glucose and lipid metabolism, improved the texture and aroma of the fermented sausage”.
In the work on the development of the recipe of fermented sausage, Bifidobacterium Animalis A12 was used.The growth activity, tolerance, and enzyme activity of B. animalis A12 and its contribution to the texture and flavour of fermented sausages were evaluated. Additionally, the sensory texture, flavour components, and amino acid nutrients during the fermentation process were assessed. Generally the work is interesting. The obtained test results can be used in the food industry. The authors used advanced measuring techniques suitable for the adopted purpose of research.
Title
The title and the aim of the study are clearly constructed.
Abstract
The abstract includes the aim of the study, methods used in the experiment and contain the principal results and conclusions.
Introduction
The introduction describes the matter of the experiment and states the problem being investigated. Authors correctly interpreted and described the significance of the results for the research. They skillfully referred to the results of other researchers. Literature references are the most current.
Materials and Methods
Generaly, the data is well collected. The sampling is appropriate and adequately described. Statistical analysis of measurement results has been used.
Despite this, provide more information in 2.7. Determination of pH, colour, and texture of fermented sausage.
What method was the texture tested? TPA? If so, please inform me about the operating conditions of the apparatus.
How was data acquisition rate? How was time interval between the two compression cycles?
What were the values: PPS (points per second) and trigger force?
Instrumental colour determination:
An instrumental evaluation was carried out of physical colour parameters L*, a*, b* of the of the fermented sausage samples, in CIE? The reflection method within the range of visible light? Observer settings, i.e. incidence angle, calibration parameters….
Results and Discussion
The authors correctly interpreted and described the significance of the results for the research.
Conclusion
The authors correctly indicate, how the results are related to the studies.
References
Literature references are appropriate and relate to the position from the last few years.
Language
The article is correctly written. English language and style are minor spell check required.
Author Response
We appreciate all the efforts you have to our manuscript. All the comments were very valuable to us. And had been carefully addressed in the revision. In the manuscript with tracked changes, the part marked with the red highlight was the revision following Reviewer 1’s advice. Moreover, we make some explaination to your questions.We hope our response to the comments satisfy reviewer.
Comments 1: What method was the texture tested? TPA? If so, please inform me about the operating conditions of the apparatus. How was data acquisition rate? How was time interval between the two compression cycles? What were the values: PPS (points per second) and trigger force?
Response 1: Using the texture profile analysis (TPA) method, the P50 cylindrical probe was used to cut the fermented sausage into samples with a diameter of 12.7 mm and a length of 10.0 mm. The compression rate was 2.0 mm/s before the test, and 1.0 mm/s during the test. After test, the compression rate was 1.0 mm/s, the compression ratio was 75%, and the pause time between the two compressions was 5.2 s. and the following textural parameters were calculated: hardness(N), chewiness(N*mm) and elasticity(mm). (page 4 line 155-161)
Comments 2: An instrumental evaluation was carried out of physical colour parameters L*, a*, b* of the of the fermented sausage samples, in CIE? The reflection method within the range of visible light? Observer settings, i.e. incidence angle, calibration parameters….
Response 2: The colour was measured with a colorimeter (KCM-700d, ONICA MINOLTA, Japan) using an 8 mm port size, illuminant D65, and 10â—¦ standard observer. The L*(lightness), a*(red) and b*(yellow) value of fermented sausage were measured and recorded after whiteboard correction. There were 3 parallel samples for each sample, 5 points for each parallel sample, a total of 15 measured values, and the average of 15 measured values was used as the analytical standard. (page 4 line 149-153)

Reviewer 2 Report
The article entitled “Bifidobacterium animalis A12, a probiotic strain that promotes glucose and lipid metabolism, improved the texture and aroma of the fermented sausage” shows clearly the suitability of this LAB strain to produce high-quality fermented sausages. The study provides valuable information regarding the main sensory quality parameters, as well as the strain’s ability to serve as a starter in the fermentation process.
From my point of view, this research significantly contributes to the development of improved fermented meat products with high-added value, meeting the current market demand for healthier products. Even so, I have some points that I would like to comment and that should be improved or, if the authors prefer, refuted.
Major comments
1. From my point of view, it would be appropriate to expand the abstract a little, adding a final opinion about the implication(s) that the findings of this study may have in the sector.
2. I think that everything explained in relation to how the sensory evaluation was done (lines 215-220 and table 2) should be explained in the material and methods section.
3. An experimental batch with and without A12 strain would have been interesting to see the influence of this LAB on all the study parameters.
4. In addition to presenting the most relevant data from the study, the possible implications of using A12 strain should be added at the end of the conclusions section.
Minor comments
1. In the tittle, the words “Bifidobacterium animalis” (line 2) should be italicized.
2. The same for the word “B. animalis” (line 23). Please, check that all the words referring to the genera and species of microorganisms are spelled correctly throughout the manuscript.
3. “Lactic acid bacteria”, on line 56, was already written above (line 55), so move the abbreviation “LAB” from line 56 to line 55.
4. In my opinion, the sentence “Xi et al found that [7]” (line 61) should be written as “Xi et al. [7] found that”.
5. The authors Coz and Bk (line 65) are actually Özer and Kılıç. Please, correct it both in the text and in the list of references.
6. From my point of view, the reference “Hu et al…[11]”, on line 70, would be better written as “Hu et al. [11]” at the beginning of the sentence. This is only a personal opinion and may be valid for the rest of the document.
7. On line 112, the word strain should be put in the plural, as it appears in the tittle of the previous paragraph (line 106).
8. In section 2.7. Determination of pH, colour, and texture of fermented sausage, I would like to know the type of texture experiment carried out. For instance, was it a TPA or Warner-Bratzler assay? Moreover, the evaluated texture parameters should be described, as well as the units used.
9. In section 2.8. Determination of hydrolysed amino acids, I would like to know the working conditions and the equipment used (e.g., type of column, operating flow, detector…). Something similar to what is written in section 2.9. I think it is an important and very interesting information for the reader.
10. On line 167, the parenthesis in “(Figure 1A)” should be removed”. This is extensible for the rest of the manuscript.
11. Rewrite the sentence “when the…decreased greatly” (lines 184-185), capitalizing the word “when” at the beginning and adding but or however to make sense to the phrase.
12. What country is regulation GB5009.33-2016 from? (line 193). I find it interesting to inform the reader about it.
13. The phrase “As shown in…product starter” (lines 195-199) should be rewritten because it is confusing. The beginning implies that the addition of nitrite impairs the growth of the A12 strain, which is to be expected and true, but then it is said that A12 has good qualities, being a good strain to be used in the fermentation of sausages. This is obvious by looking at figure 1D, but not by only reading the text. I think the authors should explain this better.
14. In the sentence “a, b, c…(P<0.05)” (lines 202-203) the word nitrite should not also be added? The phrase as written is correct for figure 1C but not for figure 1D.
15. Indicate at the bottom of table 3 the meaning of the letters and the significance value, as was done for figure 1.
16. Change the coma to a dot in “[29], Among them,” (lines 271-272).
17. The same for “[32], at the” (line 274).
18. Italicize “Bifidobacterium animalis” on line 291.
19. I notice that no letters were added to the pH row at different times in Table 4. I suppose that no significant differences were found along the fermentation time. However, no differences were found in the color traits either and a letter was added. Please, unify criteria.
20. In the elasticity row of Table 4 the letters c and d are missing.
21. In figure 3, it would be nice to include the units for the number of amino acids (y-axis).
22. The phrase “Methylheptenone…fragrance” (lines 363-364) needs to be rewritten because it is not well-understood.
23. In figure 4, the units should be added to the amount of volatile compounds (y-axis).
24. Put “Debaromyces hansenii” in italics (line 486). Check the rest of the references as this type of error was found several times throughout the section.
25. Remove “[19]” on line 525.
26. The same on line 573.
27. Remove “[j]” on line 576.
28. Check the citation format (lines 651-652).
Author Response
Thank you so much for your detailed and comprehesive advice. In the revision manuscript with tracked changes, the part marked with the yellow highlight was the revision following Reviewer 2’s advice. Moreover, we make some explaination to your questions.We hope our response to the comments satisfy reviewer.
The number of page and line shown in “response” is marked in the manuscript with Tracked Changes.
Major comments
Comments 1: From my point of view, it would be appropriate to expand the abstract a little, adding a final opinion about the implication(s) that the findings of this study may have in the sector.
Response 1: Thanks very much for your valuable comments, we have added a final opinion about the implication(s) that the findings of this study may have in the sector. (page 4 line 30-31)
Comments 2: I think that everything explained in relation to how the sensory evaluation was done (lines 215-220 and table 2) should be explained in the material and methods section.
Response 2: We think the comment are very desirable, we have made changes based on the comments. (page 3 line136-143)
Comments 3: An experimental batch with and without A12 strain would have been interesting to see the influence of this LAB on all the study parameters.
Response 3: Thanks very much for your valuable comments, we will further explore the effect of with or without A12 on all the study parameters in the future.
Comments 4: In addition to presenting the most relevant data from the study, the possible implications of using A12 strain should be added at the end of the conclusions section.
Response 4: We think the comment are very desirable, we have added the implications of using A12 strain at the end of the conclusions section. (page 15 line 493- 495)
Minor comments
Comments 1: In the tittle, the words “Bifidobacterium animalis” (line 2) should be italicized.
Response 1: Thanks for your advice, we have italicized the words “Bifidobacterium animalis”. (page 1 line 2)
Comments 2: The same for the word “B. animalis” (line 23). Please, check that all the words referring to the genera and species of microorganisms are spelled correctly throughout the manuscript.
Response 2: Once again, We are very sorry for this oversight, and we checked all words involving microbial genera and species, which were spelled correctly throughout the manuscript. (page 1 line 24)
Comments 3: “Lactic acid bacteria”, on line 56, was already written above (line 55), so move the abbreviation “LAB” from line 56 to line 55.
Response 3: Thanks for your advice. We have modified it in the manuscript. (page 2 line 58, 59)
Comments 4: In my opinion, the sentence “Xi et al found that [7]” (line 61) should be written as “Xi et al. [7] found that”.
Response 4: According to your comment, We have modified it in the manuscript. (page 2 line 64)
Comments 5: The authors Coz and Bk (line 65) are actually Özer and Kılıç. Please, correct it both in the text and in the list of references.
Response 5: According to your comment, We have modified it in the text and in the list of references. (page 2 line 69; page 9 line 532)
Comments 6: From my point of view, the reference “Hu et al…[11]”, on line 70, would be better written as “Hu et al. [11]” at the beginning of the sentence. This is only a personal opinion and may be valid for the rest of the document.
Response 6: Thanks for your advice, we have modified in the manuscript. (page 2 line 74, 76)
Comments 7: On line 112, the word strain should be put in the plural, as it appears in the tittle of the previous paragraph (line 106).
Response 7: As suggested by the Reviewer, we have modified in the manuscript. (page 3 line 117)
Comments 8: In section 2.7. Determination of pH, colour, and texture of fermented sausage, I would like to know the type of texture experiment carried out. For instance, was it a TPA or Warner-Bratzler assay? Moreover, the evaluated texture parameters should be described, as well as the units used.
Response 8: Using the texture profile analysis (TPA) method, the P50 cylindrical probe was used to cut the fermented sausage into samples with a diameter of 12.7 mm and a length of 20.0 mm. The compression rate was 2.0 mm/s before the test, and 1.0 mm/s during the test. After test, the compression rate was 1.0 mm/s, the compression ratio was 75%, and the pause time between the two compressions was 5.2 s. and the following textural parameters were calculated: hardness(N), chewiness(N*mm) and elasticity(mm). (page 4 line 155-161)
Comments 9: In section 2.8. Determination of hydrolysed amino acids, I would like to know the working conditions and the equipment used (e.g., type of column, operating flow, detector…). Something similar to what is written in section 2.9. I think it is an important and very interesting information for the reader.
Response 9: Instrument conditions: Analytical column: 4.6mm×60mm; Resin: 2622#; Column temperature: 57℃; Reaction column temperature: 135℃; Buffer: citric acid, sodium citrate buffer; Chromogenic solution: ninhydrin solution. (page 4 line 169-171)
Comments 10: On line 167, the parenthesis in “(Figure 1A)” should be removed”. This is extensible for the rest of the manuscript.
Response 10: According to your comment, we have removed of the parenthesis in “(Figure 1A)”. (page 5 line 196)
Comments 11: Rewrite the sentence “when the…decreased greatly” (lines 184-185), capitalizing the word “when” at the beginning and adding but or however to make sense to the phrase.
Response 11: According to your comment, we have rewritten the sentence as “When the NaCl concentration was 0-2.5%, All three strains had high OD600, the NaCl concentration was 5%, the OD600 of S2 and T1 significantly decreased (P<0.05)”. (page 5 line 213-215)
Comments 12: What country is regulation GB5009.33-2016 from? (line 193). I find it interesting to inform the reader about it.
Response 12: GB5009.33-2016 regulation from China. (page 5 line 223)
Comments 13: The phrase “As shown in…product starter” (lines 195-199) should be rewritten because it is confusing. The beginning implies that the addition of nitrite impairs the growth of the A12 strain, which is to be expected and true, but then it is said that A12 has good qualities, being a good strain to be used in the fermentation of sausages. This is obvious by looking at figure 1D, but not by only reading the text. I think the authors should explain this better.
Response 13: According to your comment, we have rewritten the sentence as “In this study, according to actual needs, the addition amount was 30mg/kg, as shown in Figure 1D, at this time, the nitrite tolerance of B. animalis A12 was similar to that of commercial starter culture, and all three strains had high OD600 values, indicating that B. animalis A12 could adapt to the nitrite content in the experiment and maintain good growth ability, which could be used as meat product starter culture”. (page 5 line 224-229)
Comments 14: In the sentence “a, b, c…(P<0.05)” (lines 202-203) the word nitrite should not also be added? The phrase as written is correct for figure 1C but not for figure 1D.
Response 14: Thanks for your advice, we have added nitrite. (page 6 line 233)
Comments 15: Indicate at the bottom of table 3 the meaning of the letters and the significance value, as was done for figure 1.
Response 15: We are very sorry for this oversight, we have added table 3 the meaning of the letters and the significance value. (page 7 line 277)
Comments 16: Change the coma to a dot in “[29], Among them,” (lines 271-272).
Response 16: We have revised. (page 8 line 297)
Comments 17: The same for “[32], at the” (line 274).
Response 17: We have revised. (page 8 line 300)
Comments 18: Italicize “Bifidobacterium animalis” on line 291.
Response 18: We have italicized the words “Bifidobacterium animalis”. (page 9 line 318)
Comments 19: I notice that no letters were added to the pH row at different times in Table 4. I suppose that no significant differences were found along the fermentation time. However, no differences were found in the color traits either and a letter was added. Please, unify criteria.
Response 19: We are very sorry for this oversight, we have unified the criteria. (page 10 line 332)
Comments 20: In the elasticity row of Table 4 the letters c and d are missing.
Response 20: We have added letters c and d in the elasticity row of Table 4. (page 10 line 333)
Comments 21: In figure 3, it would be nice to include the units for the number of amino acids (y-axis).
Response 21: According to your comment, we have added the units for the number of amino acids. (page 11 line 360 Figure 3)
Comments 22: The phrase “Methylheptenone…fragrance” (lines 363-364) needs to be rewritten because it is not well-understood.
Response 22: We have re-written the phrase according to the Reviewer’s comment. (page 12 line 392-393)
Comments 23: In figure 4, the units should be added to the amount of volatile compounds (y-axis).
Response 23: According to your comment, we have added the units for the amount of volatile compounds. (page 13 line 399 Figure 4)
Comments 24: Put “Debaromyces hansenii” in italics (line 486). Check the rest of the references as this type of error was found several times throughout the section.
Response 24: Thanks for your advice. We have checked all references. (page 16 line 522, 525, 535)
Comments 25: Remove “[19]” on line 525.
Response 25: We are very sorry for this oversight, we have removed. (page 16 line 561)
Comments 26: The same on line 573.
Response 26: We have removed. (page 17 line 609)
Comments 27: Remove “[j]” on line 576.
Response 27: We have removed. (page 17 line 612)
Comments 28: Check the citation format (lines 651-652).
Response 28: Thanks for your advice. We have checked and revised the citation format. (page 18 line 687)
